# Diagnostic Accuracy of Shear Wave Elastography in Predicting Malignant Origins of Pleural Effusions in Emergency Departments

**DOI:** 10.3390/diagnostics15020225

**Published:** 2025-01-20

**Authors:** Rune Wiig Nielsen, Casper Falster, Stefan Posth, Niels Jacobsen, August Emil Licht, Rahul Bhatnagar, Christian Borbjerg Laursen

**Affiliations:** 1Odense Respiratory Research Unit (ODIN), Department of Clinical Research, Faculty of Health Sciences, University of Southern Denmark, 5000 Odense, Denmark; 2Emergency Medicine Research Unit, Department of Clinical Research, Faculty of Health Sciences, University of Southern Denmark, 5000 Odense, Denmark; 3Open Patient Exploratory Network (OPEN), Odense University Hospital, 5000 Odense, Denmark; 4Department of Respiratory Medicine, Odense University Hospital, 5000 Odense, Denmark; 5Department of Emergency Medicine, Odense University Hospital, 5000 Odense, Denmark; 6Prehospital Research Unit, Department of Clinical Research, Faculty of Health Sciences, University of Southern Denmark, 5000 Odense, Denmark; 7Academic Respiratory Unit, University of Bristol, Bristol BS8 1QU, UK; 8Respiratory Department, North Bristol NHS Trust, Bristol BS10 5NB, UK

**Keywords:** ultrasound, shear wave elastography (SWE), thoracic malignancy, pleural effusion, malignant pleural effusion (MPE)

## Abstract

**Objective**: Ultrasound is a valuable diagnostic tool in the diagnostic work-up of dyspnea and can identify even small pleural effusions. The incorporation of shear wave elastography (SWE) represents a possible tool in stratifying pleural effusions by the risk of underlying malignancy. No previous studies on ultrasound with the incorporation of SWE have been conducted in an emergency department (ED), where such stratification might have a clinical impact by hastening referrals for the diagnostic work-up of underlying malignancy. The objective of this study was to appraise the diagnostic accuracy of ultrasonographic findings associated with thoracic malignancy as well as to calculate the optimal cutoff values for SWE in this regard. **Methods**: Patients with a unilateral pleural effusion of unknown origin were included in the ED and subjected to a thoracic ultrasound (TUS) scan during their first 48 h after admittance. Two index tests were applied: (i) traditional B-mode TUS examination registering the presence of diaphragmatic nodules, pleural thickenings and other findings associated with malignancy and (ii) an SWE examination of different regions of interest. The reference test was defined as the subsequent diagnosis of malignant pleural effusion (MPE) in the three months following inclusion. **Results**: In total, 39 patients were included. The B-mode TUS index test yielded a sensitivity of 28.57% (95%CI 3.67–70.96%) and a specificity of 90.62% (95%CI 74.98–98.02%). The SWE max of the intercostal space yielded a sensitivity of 100% (95%CI 47.82–100%) and a specificity of 59.09% (95%CI 36.35–79.29%). **Conclusions**: A TUS with integrated SWE may aid in identifying MPEs and improving referrals for the diagnostic work-up of underlying malignancy. Larger, adequately powered studies are warranted.

## 1. Introduction

Pleural effusion (PE) is a common condition with an annual incidence of approximately 1.3 million in the United States of America [1]. PEs are categorized into transudates or exudates based on biochemical composition, encompassing the level of protein and lactate dehydrogenase (LD), with higher-osmolality exudates occurring due to local inflammation [1,2,3]. Malignant pleural effusions (MPEs) are an important subgroup of pleural effusions, in some studies accounting for 10–35% of all PEs [4,5,6]. In 95% of cases, MPEs originate from metastases in the pleural space and are a sign of disseminated or advanced cancer [4]. Lung and breast cancer are the most common causes, but lymphomas, genitourinary metastases and gastrointestinal metastases may also give rise to MPEs [4,7].

Thoracic ultrasound (TUS) is a key point-of-care tool in determining optimal PE management: It can rapidly help to identify possible underlying pathologies, especially malignancy; assist with choosing the most appropriate invasive diagnostic procedure and/or further patient imaging, e.g., diagnostic thoracentesis, chest X-ray or chest CT scan, and can guide what therapeutic interventions may be most suitable, such as therapeutic thoracotomy or diuretics [8,9,10,11].

A diagnosis of MPE can be more likely in the presence of pleural or diaphragmatic thickening and/or nodules. Shear wave elastography (SWE) is a relatively novel technique in TUS that can evaluate pleural thickening ‘stiffness’ [12,13]. It may help in differentiating malignant from non-malignant tissue due to the lower compliance of the former [14]. However, previous studies using SWE were conducted at specialist centers on a highly selected patient population by physicians experienced in TUS. As ultrasound is frequently applied in the emergency department (ED), where several patients with newly discovered PE of unknown origin are first identified, the utility of TUS’ ability to differentiate between malignant and benign origin should also be investigated in this setting of more unselected patients [15,16,17].

The aim of this prospective study was to appraise the diagnostic accuracy of ultrasonographic findings commonly associated with malignant pleural effusions among acutely presenting patients with unilateral PE of unknown origin as well as to calculate the optimal cutoff values for SWE in this regard. Secondly, we sought to identify subgroups of patients in whom TUS examination might be of particular benefit by considering additional lab results or baseline characteristics.

## 2. Materials and Methods

### 2.1. Study Design

This prospective observational study was conducted at the ED and Department of Respiratory Medicine at Odense University Hospital (OUH), a Danish tertiary referral center with approximately 65,000 emergency visits per year. Patients were enrolled between September 2021 and April 2022. The patients were deemed eligible for enrollment if they (I) presented as hemodynamically stable (so as to not impede critical care), (II) had no permanent cognitive disabilities, (III) were at least 18 years old and otherwise (IV) capable of providing informed consent and (V) were diagnosed with unilateral pleural effusions of unknown cause by prior imaging. Patients were included as a convenience sample, either at the ED during the first two days of admittance or from the outpatient respiratory clinic when they were referred for planned thoracocentesis.

### 2.2. Thoracic Ultrasound with Integrated Elastography—Index Test

The included patients underwent a thoracic ultrasound examination, based on the 14-zone focused lung ultrasound (FLUS) protocol previously validated for an emergency setting [18,19,20]. However, special emphasis was put on a set of B-mode findings, to assess whether they were associated with malignancy or whether an effusion is of exudative or transudative nature, see Figure 1. Earlier studies suggest a parietal pleural thickening larger than 1 cm and a nodular diaphragm thicker than 7 mm as optimal cutoff points to determine malignancy [12]. Notable findings suggestive of malignancy were regarded as lung tissue with limited/incomplete ‘fish-tail’ atelectasis despite massive effusions or varied echogenicity/heterogenicity of lung tissue not similar to typical pneumonia (hepatization of lung tissue with or without air bronchograms) [21,22,23,24]. After the B-mode scan, the patients underwent SWE, and the tissue shear wave velocity was recorded by regions of interest (ROIs) superimposed on areas specified in Figure 1.

The ultrasonographic assessment of the lung and pleura was conducted using a GE LOGIQ E9 or E10 (General Electronic, Bloomington, IL, USA) with a convex 1–6 MHz probe and an abdominal preset with crossbeam disabled. For the examination of the anterolateral zones, the patient was placed in a supine position. If feasible, the patient was placed in a seated position for the examination of posterior zones. The tissue velocity was analyzed through the ultrasound device’s SWE software. As the SWE analysis could take up to 50 s and as most patients were suffering from acute dyspnea due to their pleural effusion, the patients were not asked to hold their breath during the SWE assessment. For a thorough explanation of the procedure, see Appendix A.

Index test criteria being considered diagnostic for MPE were defined for the B-mode findings and the SWE. The B-mode ultrasound findings, as suggested by previous studies [12,13], and the SWE-based index test required post hoc analysis of the SWE results. The B-mode index test suggested malignancy if any of the following were observed: (i) diaphragmatic noduli, (ii) diaphragmatic thickening >7 mm, (iii) pleural thickening >1 cm, (iv) visible tumors or consolidations suspicious of malignancy.

Furthermore, the presence of septation and/or plankton/‘swirling sign’ was tested as a predictor of the exudative nature of a PE, as sonographic echogenicity was earlier shown to be a predictor of fluid composition [25,26].

All ultrasonographic assessment was carried out by a single operator, R.N., with a basic education in TUS, as taught in the European Respiratory Society’s TUS training program, as well as approximately 30 h of experience with the TUS protocol, corresponding approximately to that of the average physician in an ED.

### 2.3. Initial Test Result Follow-Up

The laboratory results from blood samples drawn at the initial assessment of the patient were recorded, in addition to the results of the chest X-ray and thoracocentesis, if any. For a complete list of the recorded paraclinical test results, see Appendix B.

### 2.4. Reference Test

For the reference test, two physicians, N.J. and A.L., each independently reviewed the patients’ records to determine the cause of the pleural effusion at least three months after the initial ultrasound assessment. Their assessment was based on, but not limited to, results from clinical examination, clinical imaging, biopsy, surgery, autopsy or a combination thereof. In the case of a disagreement between the two journal auditors, a third auditor would make the final decision. The auditors were blinded to the result of the index test.

### 2.5. Statistical Analysis

All the data were collected using a REDCap database and analyzed using STATA/BE 17.0. To describe the patient population, a Shapiro–Wilk’s test was used to assess the normality. Normally distributed data were reported as mean +/− standard deviation (SD) and compared with a Student’s *T*-test. Non-normally distributed data were reported as median with associated inter-quartile ranges (IQRs) and compared using a Mann–Whitney test. Categorical data were compared using Fisher’s exact test. The diagnostic accuracy of our index test was reported as sensitivity, specificity, positive and negative predictive values and as accuracy and positive and negative likelihood ratios, and their 95% confidence intervals (95% CIs) were calculated using the recommended approaches [27,28]. Numerical data for the SWE-based index test were analyzed with receiver operating characteristic (ROC) curves, and the cutoff values were examined using the Youden index.

### 2.6. Ethics

This study was conducted in accordance with the amended Declaration of Helsinki. The regional ethics board waived approval of the project (Jr. No. 21/28576). All the patients provided written informed consent prior to inclusion. The project was approved by the local branch of the Data Protection Agency prior to initiating this study. All the data were stored in a REDCap database utilizing the assistance of the Open Patient Exploratory Network (OPEN). All the data are reported in concordance with the STARD reporting guideline for diagnostic accuracy studies.

## 3. Results

A total of 49 patients were screened for inclusion during the study period (Figure 2). In total, 39 patients were included in the final study. Of the included patients, 33 were subject to SWE analysis, and, of these, 27 SWE scans of the intercostal space yielded results for post hoc analysis for an SWE-based index test (the highest of any SWE scan site).

Of the 39 total patients, 7 patients had MPE according to the three-month journal audit, with no disagreements between the auditors. The presence of malignancy was determined based on the results of the histocytological analysis from either thoracocentesis (*n* = 5) or biopsy (*n* = 2). The remaining 33 patients were deemed to suffer from benign PE, mainly due to heart disease (*n* = 8), pneumonia (*n* = 6) or due to unknown causes (*n* = 8). Baseline characteristics as well as the causes of the pleural effusion are shown in Table 1.

### 3.1. Thoracic Ultrasound Findings

The B-mode findings and their relationship with the final diagnosis and the biochemical composition of the associated effusion are shown in Table 2. The presence of a consolidation suspicious of malignancy was the only ultrasonographic finding significantly associated with malignancy (*p* = 0.028). The assessment of any other B-mode sign, such as swirling sign or septation, did not correlate either with exudates or transudates or with benign or malignant pleural effusion, and no correlation was found between the exudative nature of an effusion and a subsequent MPE diagnosis (see Appendix B, Table A1).

### 3.2. Shear Wave Elastography Findings

Of the included patients, 33 were subjected to SWE analysis. Most commonly SWE was not technically possible due to breathlessness or the acute initiation of other diagnostic work-ups such as thoracocentesis, precluding further ultrasound investigation. The baseline characteristics of patients in whom SWE was feasible are available in Table 1.

None of the SWE analyses yielded velocities for all the measured tissues, either due to practical difficulties regarding the setup of the SWE analysis or due to complications during the elastography procedure itself. A list of the SWE results is shown in Appendix B, Table A2 while a list of instances yielding no SWE data is shown in Appendix B, Table A3. The velocity of the lung consolidation beneath the pleural effusion (*n* = 22) was found to correlate the best with malignancy, with a median velocity of 0.9 m/s (IQR 0.89–0.94) in the MPE group and 1.38 (IQR 1.24–1.53) in the benign PE group (*p* = 0.0064). The max velocity of the lung atelectasis differed as well, with 1.23 m/s (IQR 0.96–1.52) in the MPE group and 1.82 m/s (IQR 1.42–2.13) in the benign group (*p* = 0.031). The analysis that yielded the most results was the SWE of the intercostal space (*n* = 27), but the max velocity of this showed lesser statistical strength (*p* = 0.2356) in differentiating the MPE group (2.28 m/s, IQR 2.23–2.62) from the benign group (1.97 m/s, IQR 1.58–2.99). In a few instances, a pleural effusion was initially diagnosed as unilateral by prior patient imaging, but ultrasound assessment revealed a bilateral component. In such cases, the side with the larger, initially found effusion was assessed with SWE.

### 3.3. Structuring an SWE-Based Index Test

To structure a post hoc SWE-based index test, the SWE of the intercostal space was used, as this most consistently yielded velocity values for analysis (27 cases of 33). The AUROC analysis of the intercostal max velocity produced an optimal cutoff point at 2.01 m/s, with a sensitivity of 100% and a specificity of 59.09%. The area under the curve (AUC) at the cutoff point was 0.77. The ROC graph is shown in Figure 3.

The inverse relation between lung consolidation median velocity and malignancy had a sensitivity and specificity of a 100% at a cutoff point of 0.945 m/s, but the SWE process only produced results in 21 cases of consolidation analysis. Furthermore, all the patients with a lung tissue velocity <0.945 m/s had an intercostal space velocity >2.01 m/s, meaning that factoring in the lung velocity would add nothing to the test regarding sensitivity, while excluding five patients due to missing data. Hence, the SWE would deem a patient to be suffering from MPE if the tissue velocity of the of the intercostal space >2.01 m/s.

### 3.4. Diagnostic Accuracy of the Index Tests

The B-mode TUS index test correctly identified 2 cases of MPE out of a total of 7 and correctly identified 29 cases of benign PE out of a total of 34 [12,13]. This resulted in a sensitivity of 28.57% (95%CI 3.67–70.96%), a specificity of 90.62% (95%CI 74.98–98.02%), a positive likelihood ratio (PLR) of 3.05 (0.62–14.97), a negative likelihood ratio (NLR) of 0.79 (0.49–1.28) and an accuracy of 79.49% (63.54–90.70%) (Table 2). However, the few registered cases with diaphragmatic nodules and parietal pleural thickening greater than 1 cm exclusively resulted in false positives. In general, every B-mode finding was more prevalent in the benign subgroup, besides consolidations suspicious of malignancy (Appendix B, Table A4).

The SWE max of the intercostal space correctly identified all 5 cases of MPE, but only 13 of the 22 cases of benign PE. This yielded a sensitivity of 100% (95%CI 47.82–100%), a specificity of 59.09% (95%CI 36.35–79.29%), a PLR of 2.44 (95%CI 1.48–4.04) and an accuracy of 66.67% (95%CI 46.04–83.48%) (Table 3).

For a list of all SWE ROIs’ diagnostic accuracy, see Appendix B, Table A5

## 4. Discussion

The TUS is a simple bedside procedure that has previously shown promising results in the evaluation of patients with unilateral PE suspected of malignancy [12,13]. Our study suggests that a TUS examination with a focus on markers for malignancy may be utilized to stratify pleural effusions by risk of underlying malignancy, specifically in those presenting to the emergency department, and that the integration of shear wave elastography may provide some aid in the same regard. The ambitious amount of tissue regions analyzed by SWE did find several regions of interest when differentiating MPE from benign PE, but difficulties with the elastography procedure, which further reduced the sample size available for analysis, limited the significance of any findings in this regard. Furthermore, no B-mode TUS findings could predict whether the effusion was of an exudative or transudative nature, and no difference was found between the prevalence of MPEs in the two effusion types.

### 4.1. Interpretations and Perspectives

The limited diagnostic accuracy of the B-mode TUS index test suggests that this test has some potential in evaluating the risk of MPE. Qureshi et al. were, to our knowledge, the first to propose a B-mode ultrasound approach when assessing for possible MPE [12]. Qureshi et al. found 15 diaphragmatic nodules and 15 pleural thickenings >1 cm out of 52 enrolled patients, almost exclusively in their MPE subgroup, contrary to our results. Notably, Qureshi et al. included their patients through a tertiary referral center for respiratory/pleural disease, which presumably affected the prevalence of such specific sonographic findings, as most of their population was already suspected of suffering from MPE. Furthermore, the sonographic imaging was conducted by a radiologist experienced in thoracic ultrasound, which might have led to a higher diagnostic accuracy when assessing niche thoracic ultrasound findings.

Jiang et al. were possibly the first to employ SWE when assessing for MPE, by performing SWE on parietal pleural thickenings. However, the low prevalence of parietal pleural thickenings and diaphragmatic nodules in this study limited our SWE procedure’s capability to assess possible MPEs in that regard. As Jiang et al. had a study setup comparable to Qureshi et al. regarding inclusion site and sonographer seniority, this might have increased the prevalence of pleural thickenings as well. Still, of the 244 patients enrolled in their study, only 108 patients had pleural thickenings available for SWE. Jiang et al. found B-mode ultrasound results comparable to those of Qureshi et al. but increased the diagnostic accuracy with the inclusion of SWE.

When disregarding the low number of patients subjected to the analysis (*n* = 21), the SWE median cutoff of 0.945 m/s when analyzing lung consolidations was the most promising result of this study, with a sensitivity and specificity of 100%, respectively. Another study, conducted by Alhyari et al., sought to evaluate SWE’s capabilities in assessing the peripheral pulmonary consolidation’s (PPC’s) underlying pathology [29]. More than half (*n* = 48) of all the PPCs evaluated by Alhyari et al. were accompanied by PE and, as such, are somewhat akin to our lung consolidation SWE ROIs. When comparing benign PPCs to those of malignant origin, Alhyari et al. found a significant difference in tissue velocities but with a cutoff value of 2.21 m/s., where a higher tissue velocity indicated a higher risk of a malignancy. We found the opposite correlation with our low cutoff value of 0.945 m/s. This opposite correlation could be due to the SWE ROIs not actually measuring malign tissue but rather airless lung tissue distal to a possible tumor, where the composition of this consolidation varies based on the etiology. A notable methodological difference is Alhyari et al.’s decision to exclude all patients who could not hold their breath while the SWE was conducted (at least 6 s). While this was incompliant with our extensive elastography approach, conducted at multiple sites on patients with acute respiratory symptoms, an SWE of an atelectasis moving synchronously with respiration would preferably be conducted on patients holding their breath.

In contrast to the study by Alhyari et al., another study, by Quarato et al., similarly reviewed the stiffness of subpleural lesions in 190 patients [30]. It found that 102 of these patients had accompanying PEs. Quarato et al. found no significant difference in stiffness between consolidations due to primary lung carcinomas, lung metastases or pneumonia. However, when excluding pneumonias with CT-diagnosed necrosis in the subpleural lesions, pneumonia had a significantly higher shear wave velocity than malignant subpleural lesions (2.95 +/− 0.68 m/s vs. 2.6 +/− 0.54 m/s, *p* = 0.006). However, the tissue velocity of malign subpleural lesions in that study is still not comparable to that found in our study, with the median of the lung tissue max velocities being 1.23 m/s (IQR 0.96–1.52 m/s, see Appendix B, Table A2.

To our knowledge, this is the first study to assess the intercostal tissue velocity when evaluating possible malignant pleural effusion. Our results suffer from a small sample size but show a possible correlation between higher velocity and malignancy, although nowhere as clear as needed for diagnosis.

### 4.2. Strengths and Limitations

The free access to advanced hospital care in Denmark and the broad inclusion method in a generalized emergency department setting increased generalizability. Furthermore, the latter ensured that the TUS and SWE procedures would be tested in a setting in which it could potentially have a clinical impact and not in a specialized referral center in which the final diagnosis could have been found through other measures at the time of inclusion. However, the inclusion process depended on emergency department physicians to diagnose pleural effusions and inform the investigator, which partly contributed to the low sample size of this study. The latter severely limited the statistical power of the subsequent analysis.

Furthermore, while this study targeted multiple ROIs for SWE to identify any site relevant for the clinical evaluation of PEs, the SWE procedure did have some inherent inconsistencies. Even though 33 patients were subjected to an SWE of at least four different ROIs, the procedure only gave results available for analysis in as little as 13 cases, in the case of pleural effusion SWE. This might be due to the Logiq E9’s capabilities as an elastography tool, the competence of the investigator, the patients’ inability to hold their breath while conducting the SWE and some tissues simply not being reasonably assessable by SWE, such as atelectases moving synchronously with respiration and effusions attenuating propagation of external stress such as SWE [31]. For ROIs such as pleural effusion and lung consolidation, no healthy controls exist due to the inherent pathological nature of pleural effusions and due to healthy lung tissue being air-filled and thus inaccessible through sonography.

In future studies, a simpler approach to SWE, targeting fewer ROIs in a greater number of patients, would be recommended, preferably with the more consistent ROIs such as lung consolidations or intercostal space.

This could possibly lead to a combined ultrasound protocol for the quicker stratification of the risk of malignancy, when evaluating patients presenting to an ED with unilateral PE.

## 5. Conclusions

In addition to TUS, integrated elastography of the intercostal space or lung consolidations may aid in the clinical evaluation of patients presenting with possible malignant pleural effusion by stratifying patients by risk of malignancy. However, while this is the only study of its kind performed in an ER setting, the results vary greatly from those of similar studies and are of low statistical power. Further research is needed on elastography’s capability in predicting the malignant origin of pleural effusions in an ER setting.

## Figures and Tables

**Figure 1 diagnostics-15-00225-f001:**
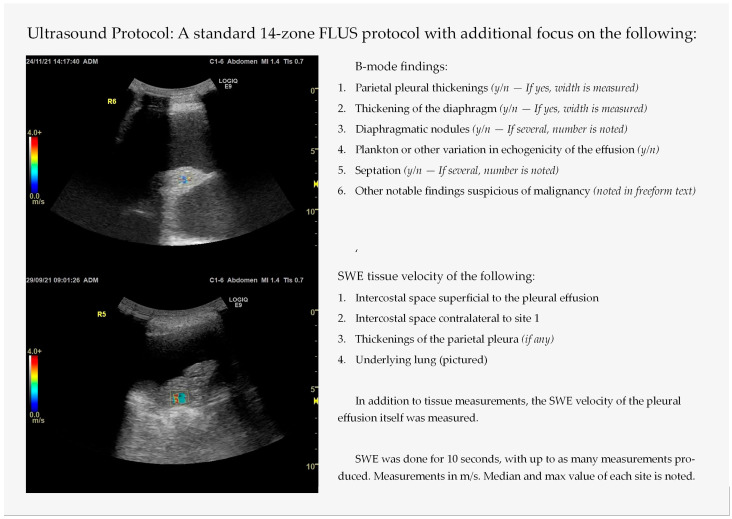
The ultrasound protocol used in this study, modified from the focused lung ultrasound (FLUS) protocol. SWE = shear wave elastography. Both sonographs pictured are of SWE of a lung atelectasis.

**Figure 2 diagnostics-15-00225-f002:**
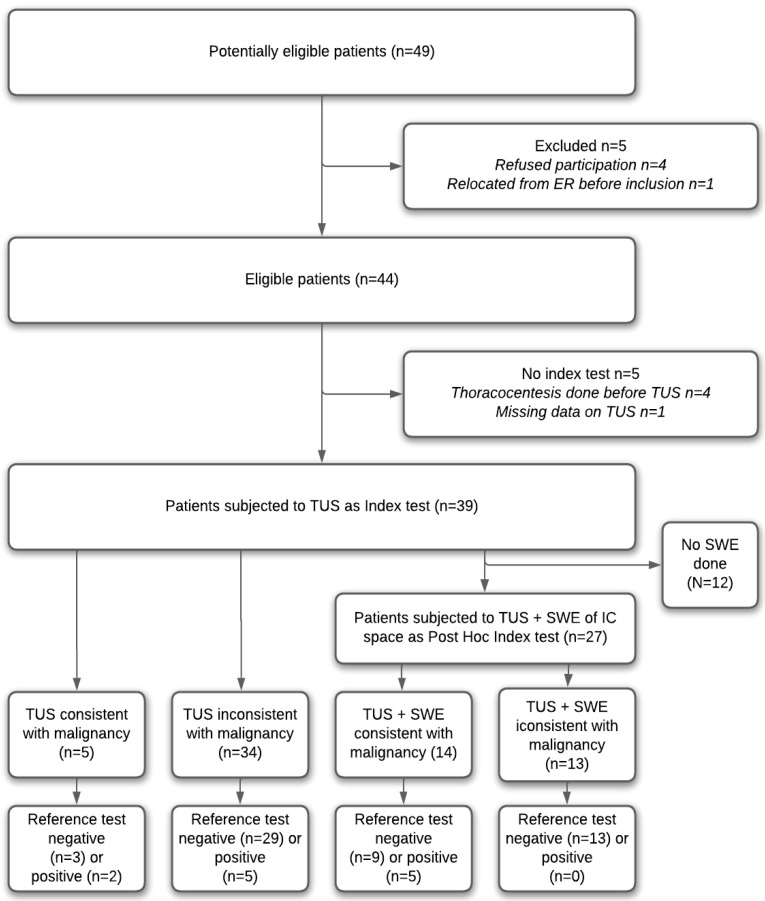
Overall study design and patient flow. ER = emergency room. TUS = thoracic ultrasound. SWE = shear wave elastography. IC = intercostal.

**Figure 3 diagnostics-15-00225-f003:**
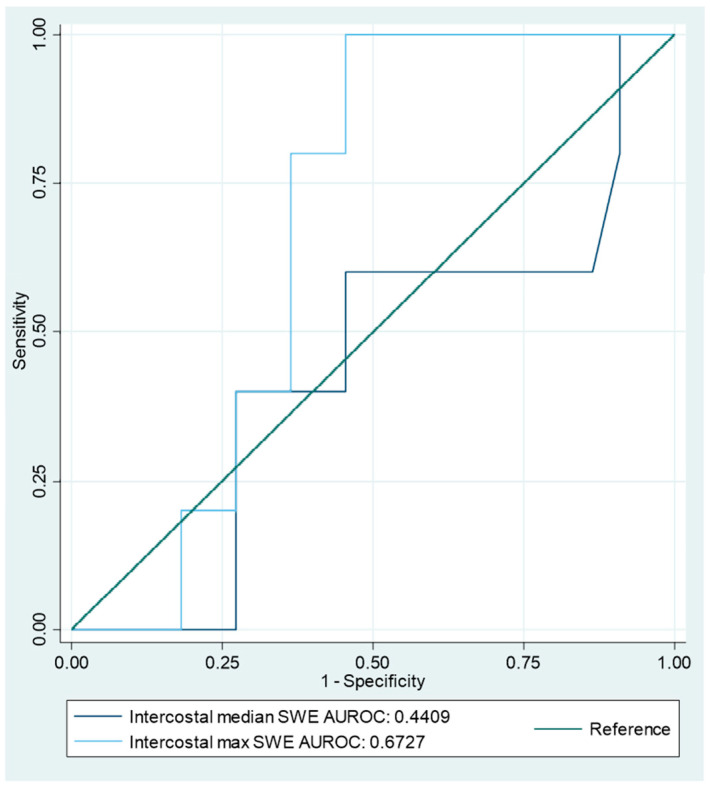
AUROC curve analysis of intercostal-space SWE. AUROC = area under receiver operating characteristic, SWE = shear wave elastography.

**Table 1 diagnostics-15-00225-t001:** Causes of pleural effusion in the study population.

	TUS	TUS + Elastography
Patients (*n*)	39	33
Mean age, years (+/−SD)	80.3 (+/−8.1)	80.72 (+/−8.43)
Male/Female	19/20	16/17
Total MPE, *n* (%)	7 (18)	6 (18.2)
Lung cancer	4 (10.3)	4 (12.1)
Breast cancer	2 (5.1)	1 (3)
Lymphoma	1 (2.6)	1 (3)
Total benign pleural effusion, *n* (%)	32 (82.1)	27 (81.8)
Pneumonia	6 (15.4)	6 (18.2)
Pleural empyema	1 (2.6)	1 (3)
Heart disease	8 (20.5)	7 (21.2)
Pulmonary embolism	2 (5.1)	2 (6.1)
Sarcoidosis	1 (2.6)	
Interstitial lung disease	1 (2.6)	1 (3)
Hepatopulmonary syndrome	1 (2.6)	1 (3)
Liver cirrhosis	1 (2.6)	1 (3)
Nephrogenic	1 (2.6)	1 (3)
Diabetic nephropathy	1 (2.6)	
Fractured rib	1 (2.6)	1 (3)
Unclear/Idiopathic	8 (20.5)	6 (18.2)

TUS = thoracic ultrasound, MPE = malignant pleural effusion, SD = standard deviation.

**Table 2 diagnostics-15-00225-t002:** Diagnostic accuracy of the B-mode index test.

B-Mode Index Test:	True Positive/Total Positive	True Negative/Total Negative	Sensitivity	Specificity	PPV	NPV	PLR	NLR	Accuracy
B-mode index test of malignancy	2/5	29/34	29% (3.7–71%)	90.6% (75–98%)	40% (12–77%)	85% (78–90%)	3.05 (0.62–15)	0.79 (0.49–1.3)	79.5% (64–91%)
B-mode index test of exudates	3/7	4/6	43% (9.9–82%)	33% (4.3–78%)	43% (21–68%)	33% (12–65%)	0.64 (0.23–1.8)	1.71 (0.47–6.3)	39% (14–68%)

Values in parentheses indicate 95% confidence intervals. PPR = positive predictive value, NPV = negative predictive value, PLR = positive likelihood ratio, NLR = negative likelihood ratio.

**Table 3 diagnostics-15-00225-t003:** Diagnostic accuracy of the SWE-based index test.

SWE-Based Index Test	Malign *n*/Benign *n*	Cutoff	Sensitivity	Specificity	AUROC	Youden	PPV	NPV	PLR	NLR	Accuracy
Intercostal velocity median	5/22	1.68 m/s	60% (15–95%)	55% (32–76%)	0.57	0.145	23% (11–41%)	86% (66–95%)	1.32 (0.56–3.1)	0.73 (0.23–2.3)	56% (35–75%)
Intercostal velocity max		2.01 m/s	100% (48–100%)	59% (36–79%)	0.77	0.545	36% (25–48%)	100%	2.44 (1.5–4)	0	67% (46–84%)
Lung consolidation velocity median	3/19	0.945 m/s	100% (29–100%)	100% (82–100%)	1	1	100%	100%	-	0	100% (85–100%)
Lung consolidation velocity max		1.525 m/s	100% (29–100%)	68% (44–87%)	0.84	0.684	33% (21–49%)	100%	3.2 (1.6–6.1)	0	73% (50–89%)

Values in parentheses indicate 95% confidence intervals. SWE = shear wave elastography, PPV = positive predictive value, NPV = negative predictive value, AUROC = area under receiver operating characteristic, PLR = positive likelihood ratio, NLR = negative likelihood ratio.

## Data Availability

The original contributions presented in this study are included in this article. Further inquiries can be directed to the corresponding author.

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
