# Peer review of "Diagnostic Accuracy of Shear Wave Elastography in Predicting Malignant Origins of Pleural Effusions in Emergency Departments"

_diagnostics, 2025, doi:10.3390/diagnostics15020225_

Round 1
Reviewer 1 Report
Comments and Suggestions for Authors
Dear Authors,
The study evaluated the diagnostic performance of SWE in pleural effusions and compared it with the performance of markers derived from B-mode ultrasound. The main finding was that B-mode ultrasound was unable to differentiate between malignant and benign origins of pleural effusions, but SWE could. The second finding was that the shear wave velocity measured in the lung consolidation was the most useful. The study is well done and the manuscript is well written. I have a few comments, but they are all minor.
1. Line 52: Please write very briefly what are the possible treatment options for PE and what/how information from TUS / SWS could be helpful in the treatment decision.
2. Line 79: It is not clear to me why hemodynamic stability and no permanent cognitive disability are necessary inclusion criteria.
3. Figure 1: Please use a better image resolution. Use the same numbering (Roman vs. Arabic numerals).
4. Figure 1: You stated to perform SWE measurements within the effusion. In a liquid-like material (e.g. cyst and blood), shear waves are instantly damped and therefore SWE measurements are not possible. This is consistent with your high failure count of 17 as reported in Table 8. So I was wondering why you chose to measure in the effusion itself. Please discuss.
5. Line 93: Since SWE is also 2D, I recommend replacing the term 2D with B-mode.
6. Line 95: To avoid misunderstanding, please clarify that you are measuring the thickening of the parietal pleural tissue and not the thickening of the pleural space.
7. Line 95: Please use a space between numbers and units (entire document)
8. Line 101: To avoid misunderstanding, please clarify that you are measuring tissue shear wave velocity, not sound velocity or tissue displacement velocity.
9. Line 107: Why do patients not have to hold their breath? This will greatly improve the quality of the SWE. How long is the required breath holding time = SWE acquisition time (without processing time)?
10. Line 119: Who was the operator? Please report the experience in TUS and SWE in hours.
11. Line 143: The cutoff is independent of AU-ROC. I think you mean "the cutoffs were examined using the Youden index".
12. Line 147: Please report the ethics number
13. All tables: Please reduce the number of (unnecessary) reported digits. For almost all numbers, you can remove all decimal places
14. Line 168: This section IS divided by subheadings
15. Figure 3: Please square the ROC plot
16. Line 275: The general observation is that malignant tissue is stiffer than benign tissue. However, you observed the opposite. Please discuss this discrepancy.
17. Line 439: Please add that the equation assumes not only isotropy, but also pure elasticity (no viscosity) and incompressibility.
18. Table 8: You stated in the caption that “2 patients were excluded from all SWE analysis due to their own wishes”. However, in the table it is reported “Patient’s wish (3)”. Please clarify.
Author Response
Thank you for your valuable comments! I thoroughly enjoyed reading your revisions, as they both added some needed clarity and forced me to elaborate on the more technical parts, which I believe an article can only gain from.
Thank your for taking your time.
I have added my answers under each comment.
- Line 52: Please write very briefly what are the possible treatment options for PE and what/how information from TUS / SWS could be helpful in the treatment decision.
-
- Thank for pointing this out. I already sought to do this in the following lines 53 – 56 but could have done so with more clarity. I have edited lines 53 - 56, I hope this adds to the manuscript.
-
- Line 79: It is not clear to me why hemodynamic stability and no permanent cognitive disability are necessary inclusion criteria.
-
- We did not include hemodynamically unstable patients, as not to impede on critical care in e.g. a trauma setting. We could not ensure that patients with cognitive disabilities were able to give informed consent, hence why they were not included.
I added a short statement regarding the former. I hope this adds clarity.
- We did not include hemodynamically unstable patients, as not to impede on critical care in e.g. a trauma setting. We could not ensure that patients with cognitive disabilities were able to give informed consent, hence why they were not included.
-
- Figure 1: Please use a better image resolution. Use the same numbering (Roman vs. Arabic numerals).
-
- I have updated Figure 1 using a better image resolution. As the original sonographs are captured on a screen with a resolution of 800 x 600 pixels, it is limited how much I can enhance the resolution on this part of the figure. However, I will attach the figure in a separate file as to ensure the highest resolution possible.
-
- Figure 1: You stated to perform SWE measurements within the effusion. In a liquid-like material (e.g. cyst and blood), shear waves are instantly damped and therefore SWE measurements are not possible. This is consistent with your high failure count of 17 as reported in Table 8. So I was wondering why you chose to measure in the effusion itself. Please discuss.
-
- This study sought to evaluate a high number of regions of interest when performing SWE, to identify any sites of clinical significance. As the viscosity of the effusions were not known before SWE, the effusions were evaluated as well. However, in hindsight, it’s obvious that the high failure rate is due to SWE’s inherent incompatibility with fluids. I’ve edited part 4.2, Strengths and limitations, to reflect this.
-
- Line 93: Since SWE is also 2D, I recommend replacing the term 2D with B-mode.
-
- Thank you for pointing this out. The original terminology adds confusion, and I’ve edited the manuscript to reflect your comment.
-
- Line 95: To avoid misunderstanding, please clarify that you are measuring the thickening of the parietal pleural tissue and not the thickening of the pleural space.
-
- As the regions of interest for SWE are specified in figure 1, I have specified this in that figure. I hope this solves any confusion.
-
- Line 95: Please use a space between numbers and units (entire document)
-
- Thank you for pointing this out, has been corrected.
-
- Line 101: To avoid misunderstanding, please clarify that you are measuring tissue shear wave velocity, not sound velocity or tissue displacement velocity.
-
- Thank you for this addition, it has been added to further clarity.
-
- Line 107: Why do patients not have to hold their breath? This will greatly improve the quality of the SWE. How long is the required breath holding time = SWE acquisition time (without processing time)?
-
- Great question! This should really have been added from the start, but I have now tried to answer this. In short, most patients admitted to the emergency department suffering from a pleural effusion were suffering from dyspnea. The SWE part of the protocol assessed up to five sites with 10 seconds of SWE of each site, yielding a possible total breath holding time of 50 seconds. We deemed this infeasible in the evaluated patient group.
-
- Line 119: Who was the operator? Please report the experience in TUS and SWE in hours.
-
- A great question. We chose an investigator with a level of TUS training approximately corresponding to that of an average ED physician, as the main objective of this study was to test the feasibility of the proposed ultrasound protocols in an average ED. The manuscript has been updated, but in short: 30 hours of TUS experience and a basic education in TUS as in accordance with the ERS TUS training programme.
-
- Line 143: The cutoff is independent of AU-ROC. I think you mean "the cutoffs were examined using the Youden index".
-
- Of course. Thank you for this clarification
-
- Line 147: Please report the ethics number
-
- Now added in the manuscript (Jr. No. 21/28576).
-
- All tables: Please reduce the number of (unnecessary) reported digits. For almost all numbers, you can remove all decimal places
-
- Thank you for this suggestion. I’ve now gone through all tables and rounded off decimals. However, I am unsure if it is sufficient, or if I perhaps have gone a little overboard.
-
- Line 168: This section IS divided by subheadings
-
- An unfortunate mistake to keep this in, thank you for the correction.
-
- Figure 3: Please square the ROC plot
-
- Now squared, thank you.
-
- Line 275: The general observation is that malignant tissue is stiffer than benign tissue. However, you observed the opposite. Please discuss this discrepancy.
-
- A great question. While we do not know for sure, I have added the following to line 274 in the discussion: “This opposite correlation could be due to the SWE ROI not actually measuring malign tissue, but rather airless lung tissue distal to a possible tumor, where the composition of this consolidation varies based on the etiology.”
-
- Line 439: Please add that the equation assumes not only isotropy, but also pure elasticity (no viscosity) and incompressibility.
-
- Of course, thank you for your insight. This has been added to line 449.
-
- Table 8: You stated in the caption that “2 patients were excluded from all SWE analysis due to their own wishes”. However, in the table it is reported “Patient’s wish (3)”. Please clarify.
-
- That is a great point! And with regards to how convoluted I now see that this table is; a great find. The explanation is rather simple: Two patients wanted to opt out completely of the SWE analysis. A third patient had their lung atelectasis assessed by SWE, but chose then to opt out. I have added “1 patient was partially excluded from SWE analysis due to their own wish” to the table text.
-
Reviewer 2 Report
Comments and Suggestions for Authors
Review for Diagnostic Accuracy of Shear Wave Elastography in Predicting Malignant Origins of Pleural Effusions in an Emergency Department by Nilesen et al:
The authors adress an important investigation in clinical practice for acute care: ultrasonography performed at initial presentation of patients with breathing problems and pleural effusions. As the nature of the diseases that lead to pleural transudates or exsudates formation can be life-threatening and generally have huge impact on morbidity and mortality, rapid recognition of the cause is of interest.
Here, the authors evaluate standard 2D thoracic ultrasound with shear wave elastography. In general, the performance of SWE is not widely performed in the emergency department, but gaining knowledge on this by the physicians could allow a rapid diagnosis in the following days after presentation.
The study is conducted in accordance with STARD criteria and the methodology is sound, providing scientific value. In the Discussion section, several studies are compared, but none was performed in the same setting and this is one of the strength of the study: the evaluation of real life diagnostic accuracy. It is obvious that the diagnostic performance is very different among the two techniques.
There are some minor aspects to adress:
1. Please discuss the implications of these findings, considering the difference between the two US methods? Could they be combined in an integrated algorythm?
2. In Key words: pleura= pleural
3. Ethics: please provide the number of approval for the study.
4. Page 6- please delete This section... lines 168-170.
Should these minor corrections be adressed, the paper should be considered for publication.
Author Response
Thank you for taking the time to go through my manuscript, I enjoyed reading your comments, and found your suggestions valuable to my manuscript.
I have added your suggestions and my answers underneath.
- Please discuss the implications of these findings, considering the difference between the two US methods? Could they be combined in an integrated algorythm?
-
- Thank you for this question. We believe that this study highlights some important regions of interest when evaluating pleural effusions with Shear Wave Elastography. However, a future algorithm or ultrasound protocol would profit from fewer regions of interest, as the current proposed protocol was lengthy and produced few usable results when comparing to the number of regions assessed. We hope that this study can point the way for future protocols to be developed, preferably focusing on consolidations or intercostal space. I have added two lines at line 321, hoping to clarify this.
-
- In Key words: pleura= pleural
-
- Edited, thank you for the correction.
-
- Ethics: please provide the number of approval for the study.
-
- Now added in the manuscript (Jr. No. 21/28576).
-
- Page 6- please delete This section... lines 168-170.
-
- An unfortunate mistake, thank you for the correction.
-
Round 2
Reviewer 1 Report
Comments and Suggestions for Authors
Dear Authors,
Thank you for the manuscript update. Almost all of my comments have been clarified/addressed to my satisfaction.
11. Line 96 and figure 1: To avoid misunderstanding, please clarify that you are measuring the thickening of the parietal pleural tissue and not the thickening of the pleural space. This has now been addressed in Figure 1 / SWS, but is still missing in Figure 1 / B-mode (1.) and line 96.
22. Line 105. Thank you for adding the spaces. Unfortunately, you missed this in line 105.
33. Line 109. If I understand you correctly, the pure acquisition time (without post-processing) of a single measurement is 10 s. This seems a bit long to me. However, the patient does not have to hold his breath for 50 seconds. The examination could be performed by holding the breath for 10 s and taking the first measurement, breathing freely, holding the breath again for 10 s and taking the second measurement, ... . Why not choose this examination protocol?
44. All tables: Regarding your concern about reducing the numbers too much: From my point of view, more could be done. For example, in Table 3 the specificity is now 54.6% (32.2% - 75.6%). The number of digits is always relative to the error or range. In your case it is from 32% to 76% and therefore over 44%. The significant digit is therefore the tens digit and indicates the rounding digit. In this case the recommended writing is 55 (32% - 76%). However, the writing as it is now will be also fine.
Author Response
Thank you, for both your valuable insight and your patience.
I have tried to address your concerns in the comments beneath.
Additionally, I have attached the revised manuscript.
- Line 96 and figure 1: To avoid misunderstanding, please clarify that you are measuring the thickening of the parietal pleural tissue and not the thickening of the pleural space. This has now been addressed in Figure 1 / SWS, but is still missing in Figure 1 / B-mode (1.) and line 96.
- Thank you for your patience. Both should have been specified now.
- Line 105. Thank you for adding the spaces. Unfortunately, you missed this in line 105.
- Has been corrected, thank you for your patience.
- Line 109. If I understand you correctly, the pure acquisition time (without post-processing) of a single measurement is 10 s. This seems a bit long to me. However, the patient does not have to hold his breath for 50 seconds. The examination could be performed by holding the breath for 10 s and taking the first measurement, breathing freely, holding the breath again for 10 s and taking the second measurement, ... . Why not choose this examination protocol?
- You are correct, the patients could have held their breath for just 10 seconds, but for a total of 4 or 5 times. We did some preliminary examinations and found 10 seconds of acquisition time to yield analyzable results more often. However, these were patients admitted to the emergency department, suffering from acute dyspnea, who voluntarily chose to participate in a study with no direct outcome on their own medical case. Some patients chose to participate in the B-mode examination but grew tired of being examined when we reached the SWE part of the examination. The mean age was 80 years, and some had a hard time keeping a sitting position. We feared that asking our patients to hold their breath for just 10 seconds would increase dropout.
- All tables: Regarding your concern about reducing the numbers too much: From my point of view, more could be done. For example, in Table 3 the specificity is now 54.6% (32.2% - 75.6%). The number of digits is always relative to the error or range. In your case it is from 32% to 76% and therefore over 44%. The significant digit is therefore the tens digit and indicates the rounding digit. In this case the recommended writing is 55 (32% - 76%). However, the writing as it is now will be also fine.
- Thank you for this explanation. This way of choosing how much to round off makes great sense, and as such, I have tried to accommodate. However, I have mainly kept it to percentages and confidence intervals, as I felt those places in the tables suffered the most from ‘number crowding’.